# Beyond Frequency Band Constraints in EEG Analysis: The Role of the Mode Decomposition in Pushing the Boundaries

Eduardo Arrufat-Pié [1,2], Mario Estévez-Báez [3,†], José Mario Estévez-Carreras [4], Gerry Leisman [2,5,*], Calixto Machado [3] and Carlos Beltrán-León [3]

1 Department of Neuroscience, Cuban Sport Research Center, Havana 10600, Cuba
2 Department of Neurology, University of the Medical Sciences of Havana, Havana 11600, Cuba
3 Department of Clinical Neurophysiology, Institute of Neurology and Neurosurgery, Havana 10400, Cuba
4 Laboratory of Neurophysiology, University Hospital Dr. Luis Díaz Soto, Havana 10900, Cuba
5 Movement and Cognition Laboratory, Department of Physical Therapy, University of Haifa, Haifa 3498838, Israel
* Correspondence: g.leisman@alumni.manchester.ac.uk; Tel.: +972-52-420-5643
† Deceased.

**Abstract:** This study investigates the use of empirical mode decomposition (EMD) to extract intrinsic mode functions (IMFs) for the spectral analysis of EEG signals in healthy individuals and its possible biological interpretations. Unlike traditional EEG analysis, this approach does not require the establishment of arbitrary band limits. The study uses a multivariate EMD algorithm (APIT-MEMD) to extract IMFs from the EEG signals of 34 healthy volunteers. The first six IMFs are analyzed using two different methods, based on FFT and HHT, and the results compared using the ANOVA test and the Bland–Altman method for agreement test. The outcomes show that the frequency values of the first six IMFs fall within the range of classic EEG bands (1.72–52.4 Hz). Although there was a lack of agreement in the mean weighted frequency values of the first three IMFs between the two methods (>3 Hz), both methods showed similar results for power spectral density (<5% normalized units, %, of power spectral density). The HHT method is found to have better frequency resolution than APIT-MEMD associated with FTT that produce less overlapping between IMF3 and 4 ($p = 0.0046$) and it is recommended for analyzing the spectral properties of IMFs. The study concludes that the HHT method could help to avoid the assumption of strict frequency band limits, and that the potential impact of EEG physiological phenomenon on mode-mixing interpretation, particularly for the alpha and theta ranges, must be considered in future research.

**Keywords:** empirical mode decomposition; Hilbert transform; Hilbert–Huang transform; EEG; non-stationary analysis





## 1. Introduction

Brain bioelectric activity exhibits oscillations with frequencies ranging from very slow, with periods of several minutes or seconds, to ultra-high frequencies between 200 and 600 Hz [1–4]. Several frequency ranges, commonly referred to as bands, have been associated with specific physiological functional states and, in some cases, brain pathologies [5]. Due to the technological limitations and constraints of bio-amplifiers, the focus of research has been on the 0.5 to 30 Hz range of the electroencephalogram (EEG) since Hans Berger's seminal publication [6], in which he reported the presence of EEG oscillations recorded from the human scalp and named them alpha and beta rhythms.

Subsequently, other bands (theta, delta, and gamma) were found to have definite physiological or pathological significance, and multiple rhythms and EEG features were described. With the development of improved bio-amplifiers, EEG oscillations with frequencies higher than 30 Hz have been studied, forming the so-called gamma band which is of high interest for cognitive and psychiatric studies [7,8]. Well-established and routinely

measured categories of ultra-fast EEG signals in the 500–1000 Hz range are brain stem evoked potentials. The study of the sources of these oscillations in different cortical and subcortical brain structures has evolved in parallel with the study of EEG frequency bands and rhythms [9–12]. Various studies have shown that when frequency bands are arranged in increasing order of their center frequencies, some degree of overlap can be observed between adjacent bands' oscillations and that the natural logarithm of these frequencies forms a linearly increasing continuum [9–13].

A key tool for studying brain rhythms was the development of the FFT algorithm [14], which was incorporated into EEG spectral analysis and continues to be used today with several improvements. The dynamics of EEG have nonlinear and non-stationary properties [15–18], which means that traditional FFT methods may not be suitable for analyzing this type of data and may result in misinterpretation of results obtained through FFT.

An innovative data-driven technique called empirical mode decomposition (EMD) has been introduced to analyze non-Gaussian, nonlinear, and non-stationary signals [19]. When combined with the Hilbert transform of the EMD-extracted modal components, this method is also known as the Hilbert–Huang transform (HHT) method and has been applied in the analysis of EEG signals [5,15,20–35]. As feature extraction of the EEG is improved through HHT, applications for other fields, such as brain–computer interface and machine learning, have arisen as a natural consequence [36–39]; however, less attention has been paid to the biological meaning of the IMFs themselves or how the mode-mixing problem affects their interpretation in terms of physiology.

In recent years, the study of the spectral frequency and power content of EEG oscillatory modes extracted through EMD has been applied for various purposes. Despite the extensive use of EMD for spectral analysis of EEG signals, there remains a lack of comprehensive studies focusing on healthy individuals and the biological interpretations of the extracted intrinsic mode functions (IMFs). Most existing studies have utilized clinical databases or have involved only a limited number of participants and EEG leads. Furthermore, the choice between the FFT and Hilbert Transform methods for IMF-based spectral analysis of EEG signals lacks clear evidence and consensus [30,40].

Given EMD's ability to decompose a signal into its natural oscillatory components [41–43], it should be possible to explore the natural spectrum of EEG without pre-established strict limits for EEG bands. This knowledge will have the potential to benefit various domains, including clinical neurology, critical patient care, cognitive sciences, brain–computer interface development, and machine learning algorithms for EEG data analysis.

This study aims to address these gaps by providing a detailed investigation of the spectral frequency and power content of different component oscillatory modes in EEG signals of healthy individuals. Two specific methods, namely FFT and HHT, will be employed, compared, and evaluated using quantitative EEG indices to assess their agreement. Moreover, this study seeks to identify potential biological interpretations of the intrinsic mode functions extracted from the EEG signals. By exploring these aspects, the study intends to contribute novel insights into the application of EMD for EEG analysis in healthy populations, thereby advancing the understanding of brain dynamics and setting a solid foundation for its future application in medical decision-making.

## 2. Materials and Methods

### 2.1. Participants and General Experimental Profile

Thirty-four healthy right-handed volunteers were included in this study, eighteen of whom were women. The inclusion criteria required participants to be over 18 years old and to have voluntarily chosen to participate. Other demographic and vital indices were measured just prior to the EEG recording. The mean age of the participants was 33.59 years with a range from 18.7 to 57.6. The mean respiratory rate per minute (RPM) was 15.76 RPM with a standard deviation of $\pm 1.7$, the mean heart rate per minute (BPM) was 68.53 BPM $\pm 9.3$, the mean systolic blood pressure was 115.21 mmHg $\pm 14.2$, and the mean diastolic systolic blood pressure was 72.62 mmHg $\pm 9.1$. The blood pressure was measured,

as well as all vital signs, using a Doctus VII, a medical monitor from COMBIOMED, a Cuban medical technology company. Participants who reported any diseases or health conditions that could modify the EEG or who had previously experienced affected awareness without a clear diagnosis were excluded. In addition, individuals who reported any toxic habits such as alcoholism, smoking, or psychotropic drug consumption were also excluded. The volunteers were studied in a laboratory with controlled temperature ranging from 22 to 24 °C, with noise attenuation and dimmed lights. A trained technician was present during the recording sessions, as well as a member of the research team (EAP).

The Ethics Committee of the Institute of Neurology and Neurosurgery, Havana, Cuba approved this research on April 2020. This study was conducted in accordance with the ethical principles outlined in the Declaration of Helsinki. The Declaration of Helsinki served as a guideline for conducting this research which involved human participants to protect their rights, safety, and well-being. The study protocol adhered to these principles by ensuring voluntary participation, informed consent, privacy, and confidentiality. In Figure 1, a block diagram of general processing steps is shown.

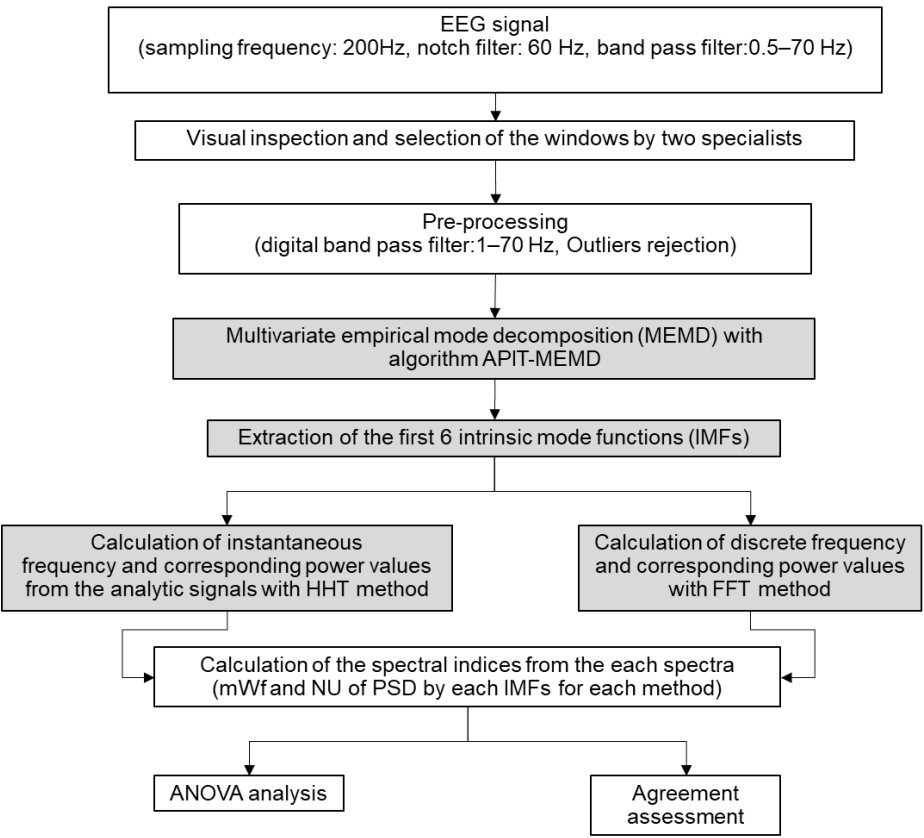

**Figure 1.** Block diagram of processing main steps. mWf_ mean weighted frequencies; NU of PSD— normalized (%) power spectral density.

### 2.2. EEG Recordings

EEG was recorded from 19 standard locations over the scalp according to the 10–20 system: Fp1, Fp2, F3, F4, C3, C4, P3, P4, O1, O2, F7, F8, T3, T4, T5, T6, Fz, Cz, and Pz. Silver scalp electrodes were fixed after careful cleaning of the skin, using a conductor paste, and connected to the input box of the digital EEG system (Medicid-05, Neuronic, S.A.: Havana, Cuba). Monopolar leads were employed, using linked ears as a reference. EEG technical parameters were: gain of 20,000, pass-band integrated analog/digital filters at 0.5 Hz and 70 Hz, "notch" filter at 60 Hz, a noise level of 2 μV (root mean square), a sampling frequency of 200 Hz, A/D converter of 15 bits, and electrode–skin impedance never higher than 5 KΩ. The records were visually inspected by two neurophysiologists to

select continuous, free-of-artifacts EEG segments with a total duration of no less than 60 s, which were later exported to an ASCII file for further quantitative analysis.

### 2.3. EEG Pre-Processing

The calculations for all results were carried out using MATLAB R2019b (version 9.7.0.1190202) internal functions, as well as customized scripts and functions created by the authors. A digital bandpass finite impulse response (FIR) filter was applied to the EEG 60 s segments digitally stored. The filter was designed with a filter order of 80 and cutoff frequencies of 1 Hz and 70 Hz to pass frequencies within the desired range. The filter was designed to have stopband attenuations of 60 dB and a passband ripple of 0.2. A sample rate of 200 Hz was considered, and the filter design method used was constrained least-squares. The required precautions were taken to avoid phase distortion in the filtering processes. To address outliers in the filtered data, the signals underwent statistical processing, filling the outliers with a piecewise cubic spline interpolation method and a moving median window of 100 sample size.

### 2.4. Empirical Mode Decomposition (EMD)

The EMD is a data-driven method for analyzing non-Gaussian, nonlinear, and non-stationary signals [19]. It is used for decomposing a signal into intrinsic mode functions (IMFs), denoted by $m_k(t)$ (k = 1 . . . K), reflecting its natural oscillations where the total number of IMFs generated for the signal is represented by K. The process of "sifting" is used to extract IMFs that have two properties: an equal number of extrema as zero crossings or differing by one, and a mean value of zero at any point for the envelope defined by local maxima and minima. The original signal X(t) is decomposed as a linear combination of IMFs and the whole process is repeated until the signal becomes a constant, monotonic function, or with a single extremum. The signal X(t) can be expressed as:

$$X(t) = \sum_{k=1}^{K} m_k(t) + f(t); \tag{1}$$

where $m_k(t)$ represents the IMFs and f(t) is the residual component [19].

### 2.5. Multivariate Empirical Mode Decomposition (MEMD)

Multivariate data, such as EEG signals, contain joint rotational modes that require coherent treatment for time–frequency estimation. In EMD, the local mean is calculated by taking an average of upper and lower envelopes, which are obtained by interpolating the local maxima and minima. However, for multivariate signals, the local maxima and minima may not be directly defined. To address this, MEMD generates multiple n-dimensional envelopes by taking signal projections along different directions in n-dimensional spaces and averaging them to obtain the local mean. This calculation is an approximation of the integral of all the envelopes along multiple directions in an n-dimensional space. The problem of finding suitable direction vectors can be treated as finding a uniform sampling scheme on an n-sphere, and a convenient choice is uniform angular sampling in an n-dimensional hyper-spherical coordinate system. A coordinate system in n-dimensional Euclidean space serves as a point set on an (n−1) sphere. Low-discrepancy sequences, such as Halton and Hammersley sequences, are used to generate a uniform point set on the n-sphere. The MEMD algorithm has been thoroughly explained by other authors [44,45].

### 2.6. Adaptive-Projection Intrinsically Transformed MEMD (APIT-MEMD)

Recently, an extension to the MEMD algorithm has been developed to counteract power imbalances and inter-channel correlations in the multichannel data of the EEG [46]. In this study, we used the APIT-MEMD algorithm for the EEG decomposition, with an alpha parameter of 0.35 and 128 directions. For calculations in the frequency domain of EEG signals, the first 6 extracted IMFs were selected, which contain the spectral range from 1 to

70 Hz as shown by other authors and our own previous experience [7,23,37,40–43,47–49]. The implementation of the MEMD and the APIT-MEMD algorithms used in this study was based on software resources publicly available at the website http://www.commsp.ee.ic.ac.uk/~mandic/research/emd.htm (accessed on 11 June 2023). The main steps of this novel algorithm are well detailed explained in the original papers [29,46].

### 2.7. Estimation of the PSD of the IMFs Using the FFT Method and Spectral Indices (FFT)

The power spectrum was calculated for each extracted IMF of every EEG lead using the Welch periodogram method, with the Matlab function "pwelch.m", a window of 512 samples, a 50% overlap, a Hamming window, and 2000 samples for the FFT. The spectral resolution was 0.1 Hz. The absolute spectral power ($P_{imf}$) of each IMF (n) was calculated for the whole-time duration of the EEG segments (60 s) from the corresponding IMFs, following the expression:

$$P_{imf_n} = \sum_{i=min}^{max} P_i;$$ (2)

where P is the discrete power values of the spectra obtained for each spectral resolution multiples (i) of each IMF (n), 'min' represents the lower limit value of the spectral range to consider (1 Hz), and 'max' is the upper limit (70 Hz in this study) [50,51].

The spectral relative power (NU) of each IMF (n) expressed in normalized units (%) was calculated as:

$$NU_{imf_n} = \frac{P_{imf_n}}{Pt_L} \times 100;$$ (3)

where $P_{imf}$ is the absolute power of each IMF (n), and Pt is the sum of the absolute power of the six IMF Welch spectra for the corresponding EEG lead (L) [50,51].

The mean weighted frequency ($mWf_{imf}$) [40] of each IMF (n) spectra, expressed in Hz, was calculated using the expression:

$$mWf_{imf_n} = \sum_{i=min}^{max} \frac{f_i \times P_i}{P_{imf_n}} ;$$ (4)

where f is the discrete spectral frequency and P is the corresponding discrete spectral power for each one of the spectral resolution multiples (i). $P_{imf}$ is the absolute power calculated for the spectrum of the corresponding IMF (n), 'min' is the lower value of the spectral range (1 Hz), and 'max' is the upper limit (70 Hz) [52,53].

For estimation of the PSD of the IMFs using the HHT method and spectral indices (HHT), the Hilbert transform was applied to each extracted IMF of every EEG, leading to the creation of analytical functions, referred to here as Z(t), that can be written as:

$$Z(t) = X(t) + iY(t);$$ (5)

where X(t) is the input time series (the IMF in this case) and iY(t) is the Hilbert transform of X(t) [19]. The instantaneous values of power and frequency can then be obtained using the following expressions:

$$P(t) = \left[ X(t)^2 + Y(t)^2 \right];$$ (6)

$$\omega(t) = \frac{d\theta(t)}{d(t)};$$ (7)

where P(t) is the instantaneous energy power, θ(t) is the phase, and ω(t) is the instantaneous frequency. Spectral indices were then calculated for the 60 s EEG segments, from which the

corresponding IMFs were extracted [19,51]. The absolute spectral power ($P_{imf}$) of each IMF (n) was calculated using the expression:

$$P_{imf_n} = \sum_{i=1}^{i=12000} IP_i;$$ (8)

where IP is the consecutive values of instantaneous power obtained from each sample (i) of the input signal; in this case, the IMF with 12,000 samples (60 s of EEG recording at 200 Hz of sampling frequency) [50]. The spectral relative power (NU) of each IMF (n) expressed in normalized units (%) was calculated as:

$$NU_{imf_n} = \frac{P_{imf_n}}{Pt_L} \times 100;$$ (9)

where $P_{imf}$ is the absolute power of each IMF (n), and Pt is the sum of the absolute power of the six IMF obtained from the corresponding EEG lead (L). The mean weighted frequency ($mWf_{imf}$) of each IMF (n) expressed in Hz was calculated, also following the recommendations of a previous author [40] using the expression:

$$mWf_{imf_n} = \sum_{i=1}^{i=12000} \frac{If_i \times IP_i}{P_{imf_n}};$$ (10)

where If represents the consecutive instantaneous frequencies and IP the corresponding values of instantaneous powers obtained from each sample (i); in this case, 12,000 samples. $P_{imf}$ is the absolute power calculated for the corresponding IMF (n).

### 2.8. Statistical Analysis

The grand average method was used to calculate the distribution of the instantaneous frequencies calculated for the IMFs in several EEG leads of interest, and of the power spectra of the IMFs calculated with the FFT method. For all the calculated indices, the normality of their distribution was explored using the Kolmogorov–Smirnov test, and the data were transformed when necessary to achieve a normal distribution before the statistical comparisons. An ANOVA test of repeated measures was carried out for the comparison of the values obtained using both methods of spectral analysis using the statistical commercial package Statistica 10 (StatSoft, Inc., Hamburg, Germany (2011). A Mann–Whitney U Test was performed to compare overlapping percentages between methods.

To analyze agreement, the normality of the distribution of the differences was tested using Shapiro–Wilk, Lilliefors, and Kolmogorov–Smirnov tests with Statistica 10 (StatSoft, Inc. 2011, Statista: Hamburg, Germany). No data transformation was needed. A Matlab script was used to perform the agreement analysis for multiple observations per individual, following the steps suggested by Bland, Altman, and Zou [54,55]. Limits of maximum acceptable differences were defined based on biologically and analytically relevant criteria [54–56], with a limit of 3 Hz and 5% for mean weighted frequencies and normalized units of power spectral density. The spectral indices were graphically represented using a modified Matlab script developed from a previous function [57]. The significance level was set at 0.05 for all tests.

### 3. Results

The multivariate empirical mode decomposition (MEMD) algorithm using APIT-MEMD revealed that the number of decomposed Intrinsic Mode Functions (IMFs) was consistent across all EEG leads in each subject. However, the power spectra of the six IMFs showed variations in magnitude, with IMF3 and IMF4 exhibiting higher magnitudes. This issue poses a challenge to the visual inspection of the spectra of all IMFs in a single diagram, as depicted in Figure 2A,B. Alternatively, representing individual IMF spectra, as shown in Figure 2A′,B′, can be a suitable solution. Although IMF7 was not considered in this study,

its mean frequency for the whole group (0.45 Hz) fell below the lower limit of the spectral range investigated (>1 Hz), and we included it in the figure for more clarity.

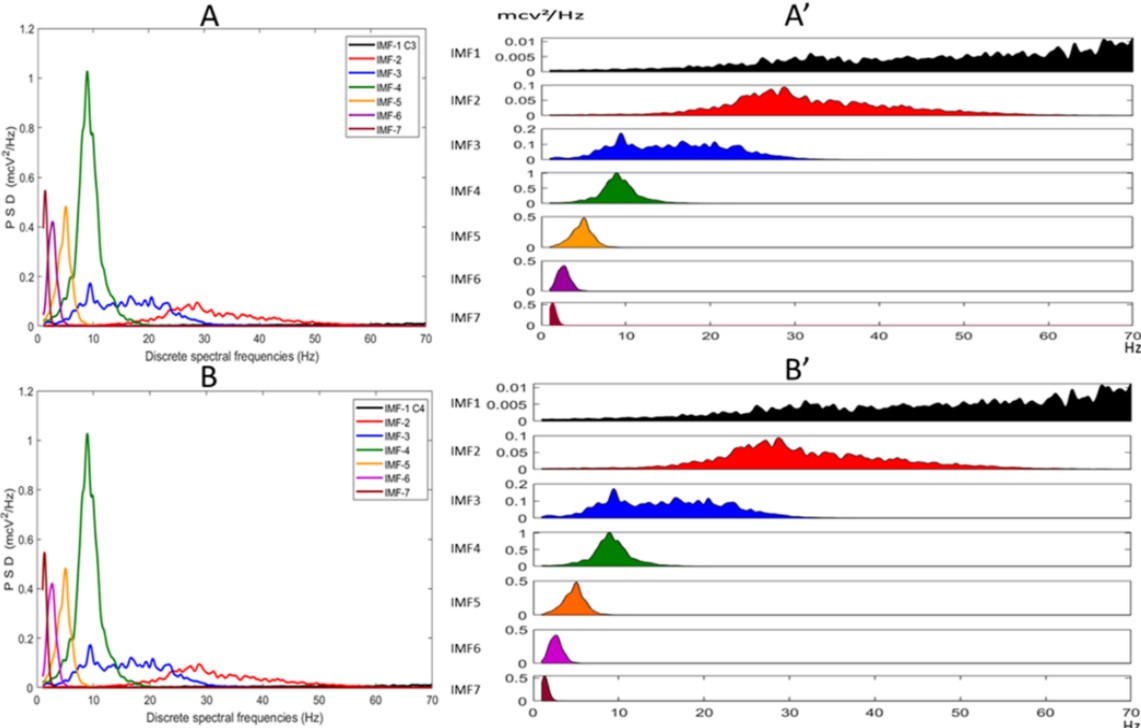

**Figure 2.** Power spectra of the IMFs extracted using the APIT-MEMD algorithm in a representative healthy volunteer. Panels (**A**,**B**) show the superimposed spectra corresponding to the EEG leads C3 and C4, while panels (**A′**,**B′**) show the same power spectra of the individual using another format that avoids the effects of the different values of the amplitude of the spectra IMFs for the visual inspection.

Regardless of the graphical method used, most of the spectra of IMFs across all EEG leads displayed overlapping spectral content between adjacent IMFs. The sequence of consecutive instantaneous values of frequency and power obtained after applying Hilbert transformation to the IMFs in one of the volunteers is presented in Figure 3 for a 2000-sample segment (10 s).

The method of grand averages showed that the overlapping of spectral content between IMFs, observed in Figure 4A,B for an individual, was also found for the whole group of healthy volunteers (see Figure 4B,C). The grand average of the distributions of the instantaneous frequencies of the IMFs showed, in general, a normal distribution for the EEG leads. An illustration of this fact is shown related to the results obtained for the EEG lead C3 (Figure 4A). Because of the constant properties of the discrete frequencies of the FFT method, only the spectra of the IMFs are depicted in Figure 4B,C. The approach based on the HHT also showed the presence of similar frequencies in different IMFs, but the degree of the overlap appears to be less marked than the observed FFT spectra of the IMFs for the same EEG lead.

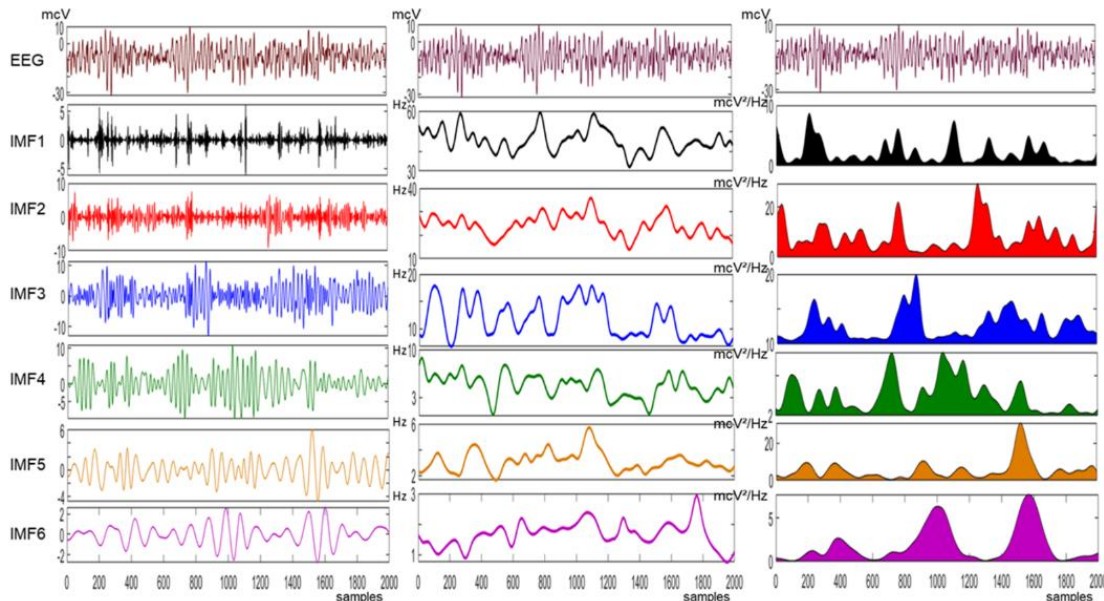

**Figure 3.** Collage showing, in the left panel, the first six IMFs decomposed from the EEG segment of a healthy volunteer using APIT−MEMD. The central panel depicts the instantaneous values of spectral frequency of the corresponding IMFs, and the right panel shows the instantaneous values of spectral power obtained using the HHT transform. The colored lines are used to differentiate the IMFs which are signaled with letters too.

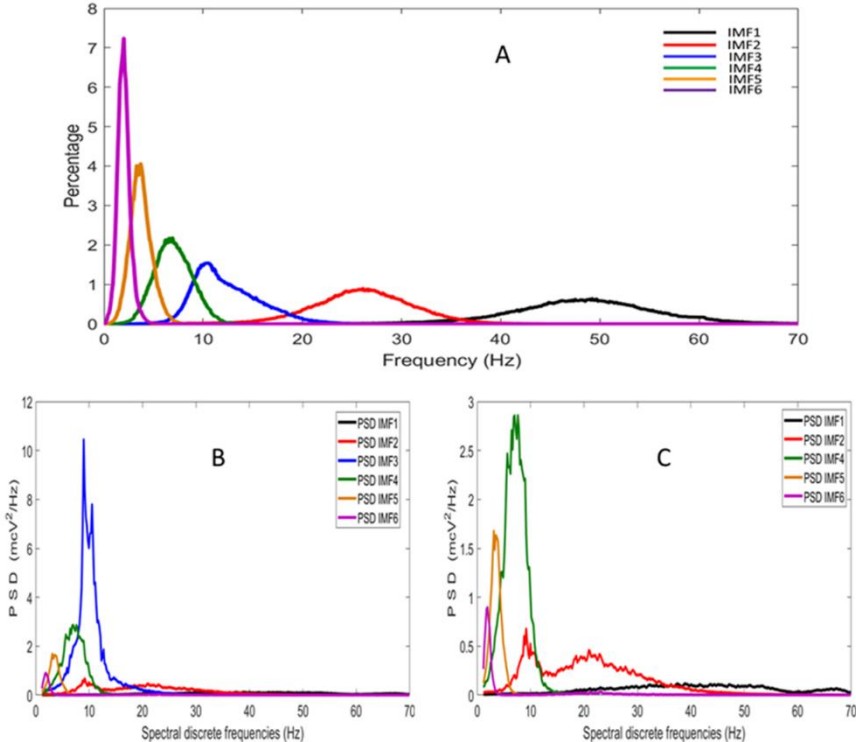

**Figure 4.** Panel (**A**) shows the grand average of distributions of the instantaneous frequency for the first 6 IMFs decomposed from the EEG of 34 healthy volunteers using the APIT-MEMD algorithm and the Hilbert–Huang Transform. Panel (**B**) depicts the grand averages of the FFT spectra of the first six IMFs, while Panel (**C**) excludes the average spectrum corresponding to IMF3 to facilitate the visual inspection of IMFs with lower PSD values (IMF1 and IMF2). Results correspond to the calculations for EEG-lead C3.

The mean weighted spectral frequencies (mWf) calculated with the two methods resulted in values that significantly differed. The HHT method showed significantly higher values for all the EEG leads of IMF 1 to IMF3, and lower values for IMF6. There were no found differences for the values calculated for IMF4 and only two for IMF5 (See Figure 5). Only some degree of the overlapping of spectral frequencies between IMF3 and IMF4 was detected for both methods. However, the values corresponding to the estimations of the PSD in normalized units (%) only showed isolated differences in some EEG leads (refer to Figure 6).

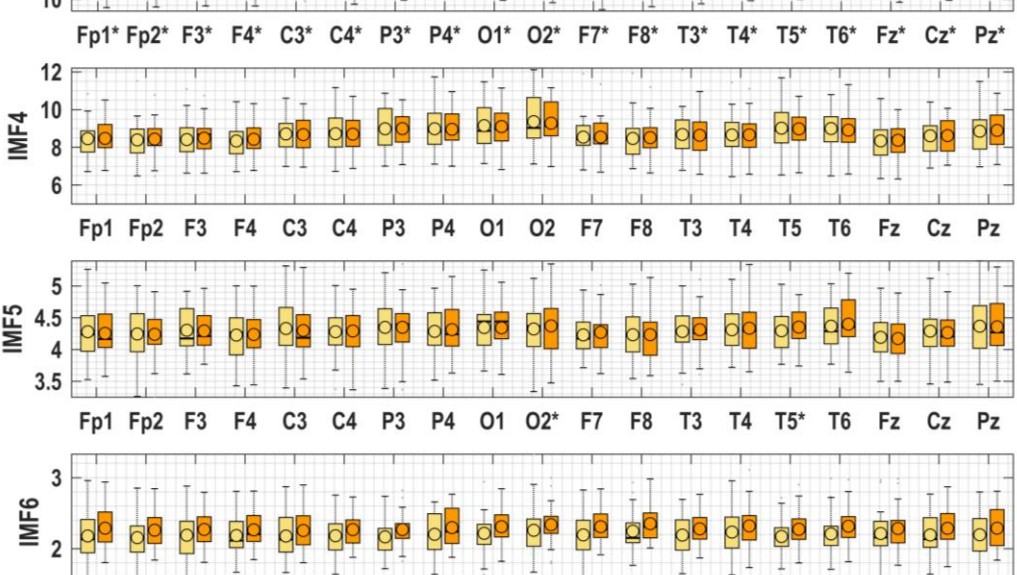

**Figure 5.** Comparison of mean weighted frequency values (Hz) between methods for each IMF from the EEG of 47 healthy volunteers. The indices were calculated using spectral methods based on the Hilbert–Huang method (yellow) and Fast Fourier Transform method (orange). Significant differences between methods are indicated by (*) for a *p* value < 0.05 in ANOVA "F" statistics. For every lead depicted, the box represents interquartile range, whiskers were calculated as 1.5× interquartile range, and circle represents data mean. "IMF" refers to intrinsic mode function.

The grand average ranges of the mWf considering both method values depicted in Figure 6 revealed that IMF1 showed values from 40.3–52.4 Hz, IMF2 showed values of 23.84–31.03 Hz, IMF5 showed values from 3.56–5.04 Hz, and IMF6 showed values 1.72–2.77 Hz, but for IMF3, values were observed from 10.01–15.79 Hz, while for IMF4 the observed values ranged between 6.74–10.71 Hz and both frequency ranges (IMF3 and IMF4) were slightly overlapped.

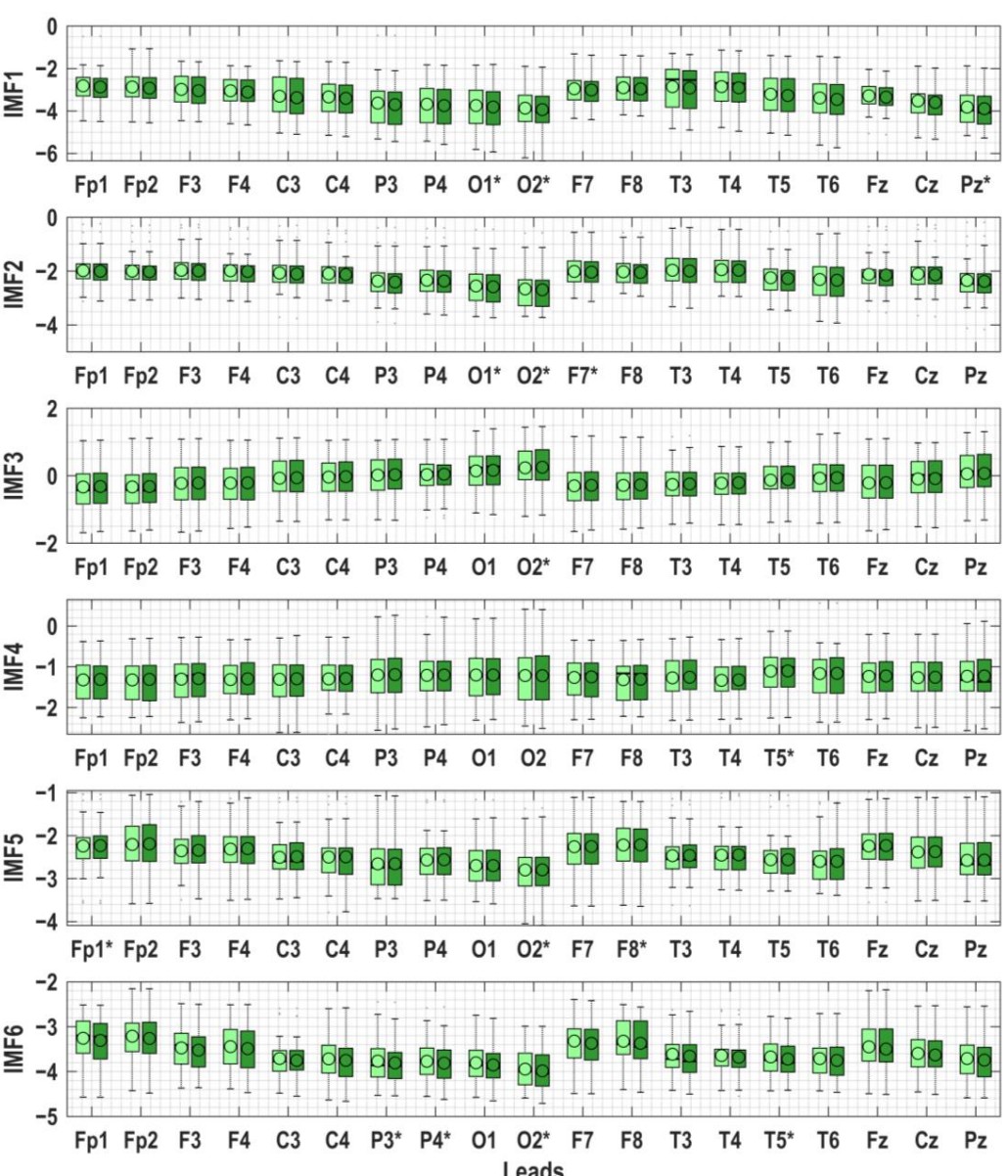

**Figure 6.** Comparison of normalized units (%) of power spectral density between methods for each IMF from the EEG of 47 healthy volunteers. The indices were calculated using spectral methods based on the Hilbert—Huang method (light green) and Fast Fourier Transform method (dark green). Normalized units (%) of power spectra were shown in logit transformation ($\log(x/1 - x)$). Significant differences between methods were indicated by (*) for a *p* value < 0.05 in ANOVA "F" statistics. For every lead depicted, the box represents interquartile range, whiskers were calculated as $1.5 \times$ interquartile range, and circle represents data mean. "IMF" refers to intrinsic mode function.

A more precise analysis of the overlapping for all groups between IMF3 and IMF4 considering the range for the mWf values (mean $\pm$ 1.96$\times$ standard deviation) for each IMF is shown in Figure 7. The spectral analysis of the IMFs with the FFT method produced more overlapping between IMFs than the HHT method.

Despite the statistical differences obtained through the ANOVA test, the agreement analysis between methods shows that only differences in mWf values on IMF1, IMF2, and IMF3 are relevant for clinical purposes (see Table 1 and Figure 8). The IMF1 showed a bias of 5.83 Hz and a 95% distribution range of 9.0 to 2.6, while IMF2 depicted a 3.42 Hz bias with a 5.7–1.1 range. Taking into consideration our previously established limits for this analysis, another minor disagreement was detected also for the range of the IMF-3 [3.3/$-$1.0]; however, the bias (1.15) was acceptable.

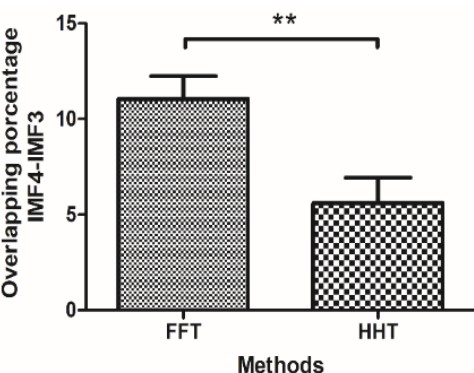

**Figure 7.** Comparison of the overlapping percentage between IMF4 and IMF 3 calculated considering the range of the 95% distribution for the mean weighted frequency values of both IMFs. Mann–Whitney U Test: (**)—$p$ = 0.0046, U = 83.50, Z = 2.83, and r = 0.46. The r values (effect size) were calculated following the expression r = Z/$\sqrt{N}$, where Z is the original Z-values previously transformed to absolute values and N is the total number of subjects submitted to the test.

**Table 1.** Agreement analysis of HHT and FFT methods for spectral indices in relaxed wakefulness with eyes-closed state *.

| Indices | IMF-1 | IMF-2 | IMF-3 | IMF-4 | IMF-5 | IMF-6 |
|---|---|---|---|---|---|---|
| | B[Ul/Ll] | B[Ul/Ll] | B[Ul/Ll] | B[Ul/Ll] | B[Ul/Ll] | B[Ul/Ll] |
| **mWf (Hz)** | 5.83 [9.0/2.6] | 3.42 [5.7/1.1] | 1.15 [3.3/$-$1.0] | 0.00 [2.0/$-$2.0] | $-$0.01 [2.0/$-$2.0] | $-$0.09 [1.9/$-$2.1] |
| **nU (%)** | 0.25 [2.2/$-$1.7] | 0.20 [2.2/$-$1.8] | $-$0.30 [1.9/$-$2.5] | $-$0.19 [1.9/$-$2.3] | 0.07 [1.9/$-$2.0] | 0.12 [2.1/$-$1.9 ] |

* Note: mWf—mean weighted frequency (Hz); nU—power spectral density expressed in normalized units (%) of the total energy; B—mean of the differences or bias; Ul—upper agreement limit; LI—lower agreement limit; B[Ul/Ll]—presentation of bias and both agreement limits in the table.

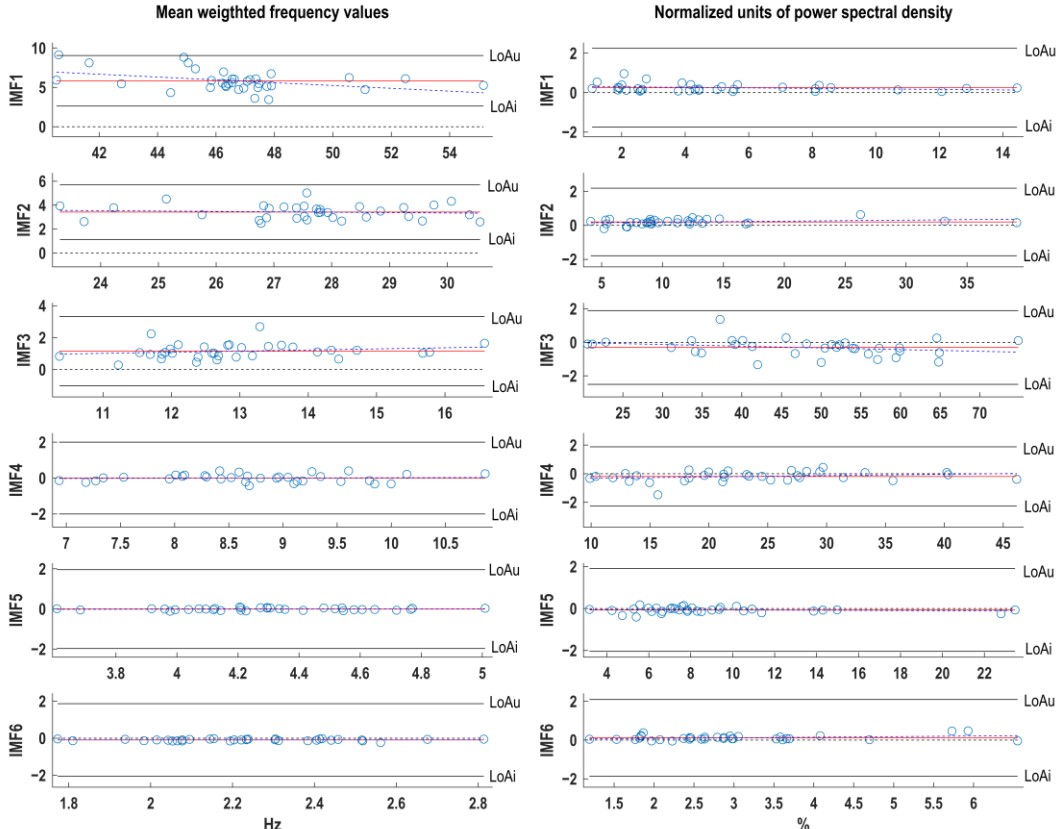

**Figure 8.** Agreement analysis between HHM and FFT methods for the mean frequency and the NU (%) of PSD values during functional resting state of relaxed wakefulness with eyes closed. The Y axes represent the differences between HHT and FFT values (HHT-FFT) and the X axes the mean of both methods (HHT + FFT/2). Red line—mean of differences; black lines—limits of the agreement (mean $\pm 1.96 \times$ standard deviation); blue discontinued line— linear trend of the differences; black discontinued line—zero level. LoAu—Upper agreement limit; LoAi—lower agreement limit; IMF—intrinsic mode function.

## 4. Discussion

Improvements in EEG feature extraction techniques using the Hilbert-Huang transform (HHT) have led to many applications in various fields, but the biological significance of IMFs and the influence of mode mixing on their interpretation have received insufficient attention, revealing limitations arising from previous assumptions about band boundaries when analyzing brain oscillators.

The results obtained using the two methods in this study showed that the first six intrinsic mode functions (IMFs) extracted from the EEG of the 19 studied scalp positions contained frequencies that approximately corresponded to the range from 1.5 to 56 Hz, considering the observed mean values of the mean weighted frequencies (mWf) (Figure 5). This range corresponds to the expected frequencies of the EEG from the classical delta to the gamma bands. The power spectral density estimates expressed in normalized units (%) of the IMFs did not show significant differences between the methods for clinical purposes considering the agreement analysis (Table 1 and Figure 8) [56]. The agreement analysis shows that a significant bias exists when the zero line (line of equality) falls outside the confidence interval. However, only biologically, analytically or clinically relevant criteria can determine whether the agreement interval is appropriate or too wide for the intended purpose. This method has previously been used to evaluate quantitative EEG methods [58] and has also been used to determine the agreement between the HHT and FFT-based methods for broadband EEG spectral analysis [53]. However, it has not been previously reported for IMF-based quantitative EEG analysis. Regardless of the results of the ANOVA

test for mWf, if we look at the individual differences between the methods for each subject, as reflected in the agreement analysis, a new result emerges, showing the first two IMFs as the major dissimilarity between the methods. The differences in the mWf can be attributed to the fact that the HHT method has a higher instantaneous frequency resolution than the FFT method, as noted by other authors [59,60], and to the fact that the Fourier transform has more constant weighted frequencies than the Hilbert transform, which allows the frequency and amplitudes to vary over time [21]. An example of this is the results obtained for EEG lead C3 (Figure 4). It is impossible to obtain a grand average of the distributions of discrete frequencies calculated using the FFT method, because these frequencies are multiples of the predefined spectral resolution and remain constant for all IMFs derived from the EEG leads of all subjects. As a result, only the FFT spectra of the IMFs were plotted (Figure 4B,C). Some authors have suggested that the first IMFs decomposed from the EEG using multivariate empirical mode decomposition could be associated with classical EEG bands [15,18,41,43]. IMF1 would correspond to the gamma band, IMF2 to the beta band, IMF3 to the alpha band, IMF4 to the theta band and IMF5 and IMF6 to sub-bands of the delta band [43].

Our results are not consistent with this. If we pay attention to the mWf shown in Figure 5, we can conclude that IMF1 for our group showed values corresponding to the gamma band, IMF2 values corresponding to the beta-2 band, IMF5 values corresponding to the theta band and IMF6 values corresponding to the delta band, but for IMF3 values of frequencies of the alpha and beta-1 bands were detected, while for IMF4 the observed values belonged to the theta and alpha bands and both frequency ranges (IMF3 and IMF4) slightly overlapped (see also supplementary material). This pattern was observed for all EEG leads, mainly for parietal and occipital leads, and was also detected for the grand averages of the distribution of the mWf (Figure 3A) and the grand averages of the spectra of the IMFs in the group of subjects for the C3 EEG lead (Figure 3 B,C). Considering that the PSD normalized values (%) for IMF3 were almost twice as high as those observed for IMF4 (Figure 4), especially for the occipital and parietal leads, where the alpha rhythm is mainly localized in healthy subjects, this allows us to speculate that IMF3 better reflects the alpha spectral frequencies and could be considered to be mainly related to this classic band, whereas IMF4 could be more associated with the theta band.

These results demonstrate one of the limitations of empirical mode decomposition, the so-called mode mixing, which consists in the appearance of different scales in an IMF, or when a signal with a similar scale appears in different IMF components. This has been reported and several methods have been proposed to mitigate its effects [18,43,47,60]. Mode mixing is also evident in the grand mean of the instantaneous frequency distributions shown in Figure 4A. Analysis of the distributions of the instantaneous frequency values is one way of assessing this problem, previously proposed by another author who analyses the mode mixing problem in a decomposition obtained from the univariate EMD algorithm [43]. Our results show that the first and second IMFs are less affected by this problem, which justifies the consideration of these components by some authors for the study of fast EEG frequencies, especially for the gamma band range [2,7,8,12,43].

The improvement of feature extraction in EEG through HHT has resulted in the emergence of applications in fields such as brain–computer interface and machine learning [36–39]. Nevertheless, the biological significance of the IMFs themselves and the influence of the mode-mixing problem on their neurophysiological interpretation has received less attention. This is a crucial problem that arises from antic assumptions, the supposition that all individuals have the same strict band limits regardless of their age, gender, or even physiological state, and the idea that the limits of all brain oscillators cannot be overlapped between themselves. The limitations of the FFT influenced these assumptions, imposed on the researchers specifically because the discrete frequency estimations are multiples of a spectral resolution and as a consequence of the Heisenberg uncertainty principle [50].

The HHT offers a possible solution: a new way to interpret the spectral properties of the EEG signal. However, the mode-mixing problem frequently interferes with this

interpretation, and more critically the EEG signal seems to be more affected than other signals, such as heart rate tachogram [53,61,62]. Another interesting piece of evidence is that mode mixing does not affect all IMFs equally; IMFs 3 and 4, related to classical alpha and theta band activities, were more overlapped than the others independently of the employed mode decomposition technique [33,51,53,63]. Important to note is the topography of the mode mixing between IMFs 3 and 4, more evident in the parietal and occipital areas, as was mentioned and depicted before (Figure 5).

In the present study, we applied different strategies to mitigate mode mixing. The EMD method that was applied to decompose the EEG was one the most novel of the multivariate versions; APIT-MEMD [46] has been developed as an improvement to MEMD [64,65], with the express purpose of coping with power imbalances and inter-channel correlations of multichannel signals, such as the EEG signal, and the method proposed by [52] was also used for the calculation of the mean spectral frequencies that mitigate the effect of some frequencies that have associated low values of power spectral density. However, as has been acknowledged by other authors, the mode-mixing problem continues to be a big issue for the analysis of multichannel signals such as EEG [17,18,47,60]. For this reason, the physiological interpretation of changes related to the third and fourth intrinsic mode functions has to take this matter into consideration. However, it must also be acknowledged that different authors have questioned the limits generally established for the EEG bands, considering the findings that several oscillatory patterns can be present simultaneously and can modulate one another [9,10,13]. Considering the frequency range and topography of the overlapping patterns, it seems that alpha rhythm variability may be a possible explanation. Our results could be evidence of this fact and this physiological phenomenon could coexist with the effect of the mode mixing, but it appears impossible to separate them.

The two methods used in this study proved to be equally informative for power spectral density estimation and the majority of the IMFs for mWf; nevertheless, careful attention must be paid to the first two IMFs. Previously, we demonstrated that the gamma band is a major point of disagreement for EEG broadband spectral analysis [56]. In this study, we offer evidence that, regardless of the decomposition process, the instantaneous frequency values offered by the Hilbert transform seem to impact positively on the high-frequency analysis and show a less overlapping percentage than the FFT method. It seems that the Hilbert transform and not the APIT-MEMD provides the most probable explanation of the major capabilities of the HHT to assess the fastest EEG frequencies. Given the previous evidence, we recommend this approach because it produces a better frequency resolution as a result of the properties of the HHT method, and also because it does not demand the existence of linearity or stationarity on the EEG record.

For this reason, this technique will be especially useful in conditions where these properties are absent and the EEG signal is truly disorganized or it varies from one person to another, such as in critically ill patients, where the previous establishment of frequencies band limits for a whole group could be a wrong assumption. Other research fields, such as brain–computer interface and machine learning, may benefit from this method, but more attention should be paid to physiological significances in those where the theta–alpha range is the main focus of investigation. The indices for broadband or narrow-band spectral analysis can be calculated from the multicomponent oscillatory modes of the EEG using both methods, and they can be interpreted by basing the analysis on their frequency ranges to assign their physiological meaning. Because of its strengths, the proposed methodology, based on an adaptive technique, could soon become an alternative to the classic spectral analysis of the EEG signals, particularly for the study of fast frequencies.

## 5. Conclusions

In this study, the Multivariate Empirical Mode Decomposition (APIT-MEMD) algorithm was used to decompose EEG signals recorded simultaneously from different locations on the scalp of healthy human subjects. This method provided distinct multi-component oscillatory modes whose frequency ranges were associated with those traditionally assigned

to spectral bands in classical electroencephalography. With the exception of frequency estimation in the first two IMFs, which contained information about faster EEG activity, the information obtained using the FFT was found to be as informative as that obtained using the Hilbert-Huang transform (HHT). Despite this similarity, HHT is recommended due to superior frequency resolution and lower overlap percentage.

Our study also observed the phenomenon of overlapping frequencies in the third and fourth oscillatory modes, a problem often reported by others authors and related to mode-mixing in EMD algorithms. However, this does not exclude the possibility that the observed frequency overlap between oscillatory modes may be due to the presence of multiple oscillatory patterns simultaneously modulating each other in the EEG. Given the frequency range and topography of the overlap, it appears that alpha rhythm variability may be related to it.

Overall, this novel approach to the study of EEG rhythms holds promise as a viable alternative to the FFT method, especially given the presence of non-Gaussian, non-linear, and non-stationary processes in EEG signals, which can lead to misleading results when using the FFT approach because it is strictly valid only for linear and stationary processes. The HHT method could also help to avoid the assumption of strict frequency band limits, but the potential impact of EEG physiological phenomena on the interpretation of mode mixing, especially for the alpha and theta ranges, needs to be considered in future research.

**Supplementary Materials:** The following supporting information can be downloaded at: https://www.mdpi.com/article/10.3390/signals4030026/s1, Table S1: Mean weighted frequency values (Hz) calculated from the intrinsic mode functions extracted from the EEG of thirty-four healthy volunteers using two spectral methods; Table S2: Power spectral density expressed in normalized units (%) calculated from the intrinsic mode functions extracted from the EEG of thirty-four healthy volunteers using two spectral metho.

**Author Contributions:** E.A.-P.: Conceptualization; data curation; formal analysis; investigation; methodology; project administration; software; resources; supervision; writing—original draft; writing—review and editing. M.E.-B.: Conceptualization; data curation; formal analysis; investigation; methodology; software; supervision; validation; visualization; writing—original draft; writing—review and editing. J.M.E.-C.: Data curation; formal analysis; investigation; methodology; validation; visualization. G.L.: Writing—review and editing; formal analysis; data curation. C.M.: Data curation; investigation; resources; supervision. C.B.-L.: Data curation; project administration; resources; supervision. All authors have read and agreed to the published version of the manuscript.

**Funding:** This research received no external funding.

**Data Availability Statement:** We regret to inform that due to institutional privacy and ethical restrictions, the raw data generated or analyzed during the course of our study is not publicly available. However, tables with the values used for the graphical representation and analysis can be checked in the supplementary material.

**Acknowledgments:** We are grateful for Mario Estevez's invaluable guidance, mentorship, and support throughout our academic journey. His knowledge and expertise in the field have shaped our research interests and provided an inspiration in the most difficult moments. Thank you for your encouragement and generosity.

**Conflicts of Interest:** The authors declare no conflict of interest.

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
