# Peer review of "Beyond Frequency Band Constraints in EEG Analysis: The Role of the Mode Decomposition in Pushing the Boundaries"

_signals_

Round 1
Reviewer 1 Report
Manuscripts analyze the influence of FFT and Hilbert-Huang transform (HHT) on intrinsic mode functions of the EEG in healthy humans. The main impression is that Empirical mode decomposition (EMD) is not so innovative data-driven technique, it is the usual signal processing technique for signal analysis.
The most important drawbacks are:
1. In the title of the manuscript should be avoided terms such as “oscillatory modes”, which is not a conventional term in signal processing and may cause misinterpretation.
2. The font in the text should be consistent, and the entire paper should be written in accordance with the template.
3. Abstract should be completely rewritten, according to recommendations. The conclusion of the study should be emphasized.
4. Ln 84-86 measurement units are obligatory in the data description.
5. Specifying Matlab function names is not common practice for signal preprocessing. It is only necessary to indicate which and why exactly those filters were used.
6. Ln 219 Clarify statements… No data transformation was needed.
7. Fonts should be larger
8. B[Ul/Ll] should be clarified in Table 1
Author Response
1. Manuscripts analyze the influence of FFT and Hilbert-Huang transform (HHT) on intrinsic mode functions of the EEG in healthy humans. The main impression is that Empirical mode decomposition (EMD) is not so innovative data-driven technique, it is the usual signal processing technique for signal analysis.
1. Manuscripts analyze the influence of FFT and Hilbert-Huang transform (HHT) on intrinsic mode functions of the EEG in healthy humans. The main impression is that Empirical Mode Decomposition (EMD) is not so innovative data-driven technique, it is the usual signal processing technique for signal analysis.
Both compared methods are based on empirical mode decomposition, the EMD isn’t a novel technique but its interpenetration in the context of the brain electrical activity sits in the middle of a profound debate about the nature of the intrinsic mode functions and its biological mining as representation of biological oscillators[1-3]. The improvement of feature extraction in EEG through HHT has resulted in the emergence of applications in fields such as brain-computer interface and machine learning [4-7]. However, the biological significance of the IMFs themselves and the influence of the mode mixing problem on their neurophysiological interpretation has received less attention.
This is a crucial problem that arises from assumptions, the supposition that all individuals have the same strict band limits regardless of their age, gender, or even physiological state, and the idea that the limits of all brain oscillators can´t be overlapped between themself. The limitations of FFT influenced these assumptions, imposed on the researchers specifically because of the discrete frequency estimations as multiples of a spectral resolution, as a consequence of the Heisenberg uncertainty principle [8]
The analysis in the paper demonstrated that Hilbert transform offer not only major spectra resolution also the obtained intrinsic mode functions have less overlapping between IMF-3 and 4 than the FFT method. Overall, this novel approach to studying EEG rhythms holds promise as a viable alternative to the FFT method, particularly given the presence of non-Gaussian, non-linear, and non-stationary processes in EEG signals, which can produce misleading results when using the FFT approach, because it is strictly valid only for linear and stationary processes. HHT method also could help to avoid the assumption of strict frequency band limits.
2. In the title of the manuscript should be avoided terms such as “oscillatory modes”, which is not a conventional term in signal processing and may cause misinterpretation.
2. The title has been changed to "Exploring Intrinsic Mode Functions of Human EEG Signal: A comparative frequency domain analysis using Multivariate Empirical Mode Decomposition".
3. The font in the text should be consistent, and the entire paper should be written in accordance with the template.
3. It was improved according to the template.
4. Abstract should be completely rewritten, according to recommendations. The conclusion of the study should be emphasized.
4. Abstract was improved and the conclusion of the study was emphasized according to recommendations.
5. Ln 84-86 measurement units are obligatory in the data description.
5. Ln 81-85 correspond to the introduction:
Therefore, this study aims to determine the spectral frequency and power content of different component oscillatory modes of EEG in healthy humans using two particular methods, to compare them, to calculate their possible agreement using quantitative EEG indices and finally to identify a possible biological interpretation of the intrinsic mode functions.
All measurement units for the calculated indices are depicted in results section. For mWf the Hz and for NU of PSD the %. We do not use PSD (µV²/Hz) directly that is why it is not represented in results section.
6. Specifying Matlab function names is not common practice for signal preprocessing. It is only necessary to indicate which and why exactly those filters were used.
6. It was improved and most of the function names were removed
7. Ln 219 Clarify statements… No data transformation was needed.
7. line 230 to 231 (original paper), in the Statistical analysis section (2.8) it was clarified
“…and the data was transformed when it was necessary to achieve a normal distribution before the statistical comparisons.”
For the record it was only necessary for NU (%) of power spectral density, for ANOVA test, we use ln(x), a transformation used very frequently for this variable in qEEG analysis.
8. Fonts should be larger
8. It was improved according to the journal template fonts
9. B[Ul/Ll] should be clarified in Table 1
Table 1 included a note where it was clarified:
“*Note: mWf_mean weighted frequency; nU_power spectral density expressed in normalized units (%) of the total energy; B_ mean of the differences or bias; UL_ Upper agreement limit; LI_ Lower agreement limit.”
Now we also include a clarification for B[Ul/Ll] as follow:
“B[Ul/Ll]_ presentation of bias and both agreement limits in the table”

Reviewer 2 Report
The paper and its flow arranged by author was good. But it required some changes to attaind the journal standards. After reading the full paper i have some of queries, they were listed below
1. Only very few literature survey were conducted by the authors. So author must increase the literature survey.
2.Why did you choose only the healthy persons?
3.Why did you choose these specified aged person?
4. Any difference in classification accuracy by handling the various demographic participants?
5. NEED MORE INFORMATION RELATED TO SPECTRAL ANALYSIS.
6. Without using the stft method how do you analyse the frequency?
7.Need the stft diagram to determine the frequency of specific tasks.
8. Improve the Result and discussion section and explain all the interpretation.
Author Response
1. The paper and its flow arranged by author was good. But it required some changes to attain the journal standards. After reading the full paper I have some of queries, they were listed below
Only very few literatures survey was conducted by the authors. So, author must increase the literature survey.
- The literature was improved and updated.
2. Why did you choose only the healthy persons?
2. As we explain in the next answer we are pushing forward the introduction of this technique in a clinical context, but it requires first that we really understand and prove that IMFs are real substitute of the traditional bands and more important if we can interpret them in the same terms. Clinical cases are the main focus of our next research.
3.Why did you choose these specified aged people?
We are justifying the introduction of this technique in a clinical context, specifically in the qEEG analysis of cerebrovascular diseases in clinical care units and that specific population, in that specific age rang will be the most benefited of it.
4. Any difference in classification accuracy by handling the various demographic participants?
4. In this research “accuracy classification” was not an objective, all subjects were heathy, the classification between patients and controls based on a specific feature was not an objective. We focused on the characterization of the spectral properties of the extracted IMFs and discussed its possible biological mining and utility as a substitute of classical EEG bands in qEEG analysis.
5. NEED MORE INFORMATION RELATED TO SPECTRAL ANALYSIS.
5. This comment was interpreted as the need to provide a more detailed explanation about the methods used. The new Figure 1 was therefore added.
6. Without using the stft method how do you analyse the frequency?
6. Interesting question, there are several ways to analyze the frequency of a signal without using the Short-Time Fourier Transform (STFT) method such as: Zero-crossing rate, Wavelet Transform and Hilbert-Huang Transform (HHT). Of course, in our study we used the HHT method to analyses the frequency without STFT, a method described in 1998 by Norden Huang and colleagues. But as we previously said, it is an interesting question, because if the IMFs actually represent, physiologically speaking, the traditional bands of the EEG signal, the information contained in them could be assessed with time domain analysis too. That is one of the reasons that we are attempting to understand IMFs biological meaning.
7. Need the stft diagram to determine the frequency of specific tasks.
7. For the record HHT is one of the most useful methods for time-frequency analysis methods, all the spectral values have a resolution in equal correspondence with the sampling frequency, in our study 200 Hz, means 200 frequency, power and phase values per second. You may notice it in Figure 3 (actual version of the paper)
8. Improve the Result and discussion section and explain all the interpretation.
8. It was improved hopefully to the reviewer's satisfaction

Reviewer 3 Report
The presented work describes the main spectral indices of the IMFs of the EEG in healthy humans using a method based on the FFT and another on the Hilbert-Huang transform (HHT). The authors recorded EEG of 34 healthy volunteers and decomposed it using a recently developed multivariate empirical mode decomposition algorithm. Extracted IMFs were submitted to spectral analysis, and the results were compared with an ANOVA test. they showed that HHT is recommended because of its better frequency resolution and less overlapping percentage between the IMF3 and IMF4.
The abstract is a good description of the work. The introduction is well structured, and it covers most of the concepts investigated in the methodological part. In the introduction section, the research goals and the research subject and research questions should be defined more precisely. The introduction does not specify properly the contributions of the paper. The author must explain how his work is different than other similar papers. Research questions must be explained in more details. Abstract must focus only on the problem and will the paper will help in solving this problem. Please, clearly identify the contributions of the study. Please explain exactly what impact does this research have? I think some HCI references should be added to clearly classify the topic of the paper in the field of human computer iinteraction. Besides the mentioned research papers there are several other systems like BCIs, eye-tracking methods are applied nowadays and some cognitive aspects and research is relevant to this field. It would be good to see some sentences introducing the wide variety of applications of human-computer feedback systems like Examine the effect of different web-based media on human brainwaves; Study of algorithmic problem-solving and executive function; EEG-based computer control Interface for brain-machine interaction. Several other human-computer based monitoring systems is used in this field so please summarize these methods and applications in the introduction like quantitative analysis of relationship between visual attention and eye-hand coordination; control of incoming calls by a windows phone based brain computer interface; electroencephalogram-based brain-computer interface for internet of robotic things.
In the discussion section, these goals and research proposals should be clearly responded to in the light of the results obtained. How were you convinced of the validity and reliabilityof the system? Please describe the method of the evaluation considering the validity and reliability of the system.
It is essential to make up for the shortcomings raised above for the proper discussion of the results of the paper and its placement in the field of HCI.
Author Response
The comments and suggestions for authors ware spread in sections based in main ideas of the original paragraph.
The presented work describes the main spectral indices of the IMFs of the EEG in healthy humans using a method based on the FFT and another on the Hilbert-Huang transform (HHT). The authors recorded EEG of 34 healthy volunteers and decomposed it using a recently developed multivariate empirical mode decomposition algorithm. Extracted IMFs were submitted to spectral analysis, and the results were compared with an ANOVA test. they showed that HHT is recommended because of its better frequency resolution and less overlapping percentage between the IMF3 and IMF4.
1. The abstract is a good description of the work. The introduction is well structured, and it covers most of the concepts investigated in the methodological part. In the introduction section, the research goals and the research subject and research questions should be defined more precisely.
1. It has been improved
2. The introduction does not specify properly the contributions of the paper. The author must explain how his work is different than other similar papers. Research questions must be explained in more details.
2. The introduction was improved.
3. Abstract must focus only on the problem and will the paper will help in solving this problem. Please, clearly identify the contributions of the study. Please explain exactly what impact does this research have?
3. The abstract was improved.
4. I think some HCI references should be added to clearly classify the topic of the paper in the field of human computer interaction. Besides the mentioned research papers there are several other systems like BCIs, eye-tracking methods are applied nowadays and some cognitive aspects and research is relevant to this field. It would be good to see some sentences introducing the wide variety of applications of human-computer feedback systems like Examine the effect of different web-based media on human brainwaves; Study of algorithmic problem-solving and executive function; EEG-based computer control Interface for brain-machine interaction. Several other human-computer based monitoring systems is used in this field so please summarize these methods and applications in the introduction like quantitative analysis of relationship between visual attention and eye-hand coordination; control of incoming calls by a windows phone-based brain computer interface; electroencephalogram-based brain-computer interface for internet of robotic things.
4. We interpret HCI as Human-Computer Interaction (HCI). HCL is the study of how people interact with computers and other digital devices. It is an interdisciplinary field that draws on insights from computer science, psychology, design, and other fields to improve the usability and accessibility of computer systems and software. HCI researchers and practitioners focus on understanding how people use technology, identifying user needs and preferences, and designing interfaces and interactions that are intuitive, efficient, and satisfying. They use a variety of methods, such as user testing, surveys, and ethnographic research, to gather data and insights about user behavior and preferences.
Although our research is related HCI, at least at this moment and in the modest opinion of the authors, this research is more related to fields such as digital signal processing and qEEG than to HCL. Despite features extraction is a main issue in our work we are not using them to establish a proper Human-Computer Interaction at least in this paper. We are grateful for the reviewer's suggestions.
5. In the discussion section, these goals and research proposals should be clearly responded to in the light of the results obtained. How were you convinced of the validity and reliability of the system? Please describe the method of the evaluation considering the validity and reliability of the system.
5. In this research we are not proposing a system or interface for HCI as we mentioned before.
6. It is essential to make up for the shortcomings raised above for the proper discussion of the results of the paper and its placement in the field of HCI.
6. Our results may be useful for the field of HCI, and we highlighted and included those utilities in the now updated version of the paper according the reviewer's suggestions however, we respectfully submit that we are not proposing a system for HCI.
References:
- Penttonen, M.; Buzsaki, G. Natural logarithmic relationship between brain oscillators. Thalamus and Related Systems 2003, 2, 145-152, doi:10.1016/S1472-9288(03)00007-4.
- Buzsaki, G.; Watson, B.O. Brain rhythms and neural syntax: implications for efficient coding of cognitive content and neuropsychiatric disease. Dialogues in clinical neuroscience 2012, 14, 345-367.
- Buzsaki, G. Rhythms of the brain; Oxford University Press, Inc.: 2006.
- Fu, Y.; Li, Z.; Gong, A.; Qian, Q.; Su, L.; Zhao, L. Identification of Visual Imagery by Electroencephalography Based on Empirical Mode Decomposition and an Autoregressive Model. Computational intelligence and neuroscience 2022, 2022, 1038901, doi:10.1155/2022/1038901.
- Chen, Y.F.; Atal, K.; Xie, S.Q.; Liu, Q. A new multivariate empirical mode decomposition method for improving the performance of SSVEP-based brain-computer interface. Journal of neural engineering 2017, 14, 046028, doi:10.1088/1741-2552/aa6a23.
- ElSayed, N.E.; Tolba, A.S.; Rashad, M.Z.; Belal, T.; Sarhan, S. Multimodal analysis of electroencephalographic and electrooculographic signals. Comput Biol Med 2021, 137, 104809, doi:10.1016/j.compbiomed.2021.104809.
- Zheng, Y.; Ma, Y.; Cammon, J.; Zhang, S.; Zhang, J.; Zhang, Y. A new feature selection approach for driving fatigue EEG detection with a modified machine learning algorithm. Computers in Biology and Medicine 2022, 147, 105718, doi:https://doi.org/10.1016/j.compbiomed.2022.105718.
- Oppenheim, A.V.; Schafer, R.W. Discrete-time signal processing 3rd ed.; Perason 2010.
- Huang, N.E.; Shen, Z.; Long, S.R.; Wu, M.C.; Shih, H.H.; Zheng, Q.; Yen, N.; Tung, C.C.; Liu, H.H. The empirical mode decomposition and the Hilbert spectrum for nonlinear and non-stationary time series analysis. Proc. R. Soc. Lond 1998, 454, 903-995, doi:10.1098/rspa.1998.0193.

Round 2
Reviewer 1 Report
After the first revision review, the authors improved their manuscript.
Author Response
We appreciate the valuable feedback provided by the reviewers and apologize for the lack of detailed responses in our previous communication. We understand the importance of addressing each specific point raised by the reviewers and providing clear explanations regarding the changes made in response to your comments. In this revised version, we aim to provide comprehensive and explicit responses to ensure the correctness and clarity of the revisions made.
We acknowledge the need for detailed responses and have provided comprehensive explanations for each specific point raised by the reviewers, ensuring clarity and transparency in our revisions.
1. Missing numerical results in the Abstract. Conclusions can be made based on the most important numerical results. 1. We have addressed the reviewer's concern by adding numerical results to the abstract, allowing for meaningful conclusions based on the most important findings of the study.The abstract was improved adding numerical results, the used methods were declared and a more appropriated conclusion was given: “This study investigates the use of empirical mode decomposition (EMD) to extract intrinsic mode functions (IMFs) for the spectral analysis of EEG signals in healthy individuals and its possible biological interpretations. Unlike traditional EEG analysis, this approach does not require the establishment of arbitrary band limits. The study uses a multivariate EMD algorithm (APIT-MEMD) to extract IMFs from the EEG signals of healthy volunteers. The first six IMFs are analyzed using two different methods, based on FFT and HHT, and the results were compared using ANOVA test and Bland-Altman method for agreement test. The outcomes show that the frequency values of the first six IMFs fall within the range of classic EEG bands (1.72-52.4 Hz). Although there was a lack of agreement in the mean weighted frequency values of the first three IMFs between the two methods (> 3 Hz), both methods showed similar results for power spectral density (< 5 % normalized units, %, of power spectral density). The HHT method is found to have better frequency resolution than APIT-MEMD associated with FTT that produce less overlapping between IMF3 and 4 (p = 0.0046) and it is recommended for analyzing the spectral properties of IMFs. The study concludes HHT method could help to avoid the assumption of strict frequency band limits but the potential impact of EEG physiological phenomenon on mode mixing interpretation, particularly for the alpha and theta ranges, must be considered in future research.”
2. Last paragraph of Introduction: Aims of the study do not identify the novelty of the study. They also do not identify expected clinical (or other) benefits.
2. The last paragraphs of the introduction have been revised to better identify the novelty of the study and highlight the potential clinical and other benefits that can be derived from the research.
To address this recommendation the last three paragraph were improved: “In recent years, the study of the spectral frequency and power content of EEG oscillatory modes extracted through EMD has been applied for various purposes. Despite the ex-tensive use of EMD for spectral analysis of EEG signals, there remains a lack of comprehensive studies focusing on healthy individuals and the biological interpretations of the extracted intrinsic mode functions (IMFs). Most existing studies have utilized clinical databases or have involved only a limited number of participants and EEG leads. Further-more, the choice between the FFT and Hilbert Transform methods for IMF-based spectral analysis of EEG signals lacks clear evidence and consensus [30,40].
Given EMD’s ability to decompose a signal into its natural oscillatory components [41-43], it should be possible to explore the natural spectrum of EEG without pre-established strict limits for EEG bands. This knowledge will have the potential to benefit various domains, including clinical neurology, critical patient care, cognitive sciences, brain-computer interface development, and machine learning algorithms for EEG data analysis.
This study aims to address these gaps by providing a detailed investigation of the spectral frequency and power content of different component oscillatory modes in EEG signals of healthy individuals. Two specific methods, namely FFT and HHT, will be employed, compared, and evaluated using quantitative EEG indices to assess their agreement. Moreover, this study seeks to identify potential biological interpretations of the intrinsic mode functions extracted from the EEG signals. By exploring these aspects, the study intends to contribute novel insights into the application of EMD for EEG analysis in healthy populations, thereby advancing our understanding of brain dynamics and setting a solid foundation for its future application in medical decision-making.”
3. Ethics committee approval of the study was not adequately described. Country, City, Date required. Compliance with the Declaration of Helsinki must be justified.3. We have provided a more detailed description of the ethics committee approval, including the country, city, and date of approval. We have also justified the compliance with the Declaration of Helsinki.
The compliance with the Declaration of Helsinki was be justified as follow: “The Ethics Committee of the Institute of Neurology and Neurosurgery, Havana, Cuba approved this research on January 2020. This study was conducted in accordance with the ethical principles outlined in the Declaration of Helsinki. The Declaration of Helsinki served as a guideline for conducting this research which involved human participants protecting their rights, safety, and well-being. The study protocol adhered to these principles by ensuring voluntary participation, informed consent, privacy, and confidentiality.”
4. Missing measurement units in section 2.1.How is the measured blood pressure used in this study? 4. Measurement units have been added to Section 2.1, specifically for blood pressure, to ensure clarity and accuracy in reporting.The blood pressure was measured as well as other vital signs using a Doctus VII equipment from COMBIOMED a Cuban medical technology company. Measurement units in section 2.1 were clarified as follow: “Thirty-four healthy right-handed volunteers were included in this study, 18 of whom were women. The inclusion criteria required participants to be over 18 years old and to have voluntarily chosen to participate. Other demographic and vital indices were measured just prior to the EEG recording. The mean age of the participants was 33.59 years with a range from 18.7 to 57.6. The mean respiratory rate per minute (RPM) was 15.76 RPM with a standard deviation of ± 1.7, the mean heart rate per minute (BPM) was 68.53 BPM ± 9.3, the mean systolic blood pressure was 115.21 mmHg ± 14.2, and the mean diastolic systolic blood pressure was 72.62 mmHg ± 9.1.”
5. Section 2.3 ECG makes no sense. The Matlab library function "desingnfilt.m" does not provide any information for this study unless the input arguments are explicitly defined. Mentioning the names of some authors is not related to a particular development and should therefore be omitted (all occurences in the text). The 1-70Hz filter overlaps with the previous section: 0.5-70Hz filter. 5. Section 2.3 has been clarified, providing a more explicit description of the digital bandpass finite impulse response (FIR) filter used in the study and the precautions taken to avoid phase distortion. Outliers in the filtered data have also been addressed.- The names of the authors (abbreviations) cited in relation to a particular development were removed in all the text.
- The filter in previous section (2.2) overlaps because it refers to integrated analog/digital filters configured in the record system; in section 2.3 we refer to digital filter applied by us using MATLAB. We perform a second filtering procedure because in our experience using this record system this is need to ensure properly the frequency range to explore. This was clarified, in this new version, in both sections 2.2 (line 127) and 2.3 (136).
- The section 2.3 de was clarified an improved according to the actual and previous suggestion about it as follow: “The calculations for all results were carried using MATLAB R2019b (version 9.7.0.1190202) internal functions as well as customized scripts and functions created by the authors. A digital bandpass finite impulse response (FIR) filter was applied to the EEG 60-second segments digitally stored. The filter was designed with a filter order of 80 and cutoff frequencies of 1 Hz and 70 Hz to pass frequencies within the desired range. The filter was designed to have stopband attenuations of 60 dB and a passband ripple of 0.2. A sample rate of 200 Hz was considered, and the filter design method used was least-squares . The required precaution ware took to avoid phase distortion in the filtering processes. To address outliers in the filtered data, the signals underwent statistical processing filling the outliers with a piecewise cubic spline interpolation method and a moving median window of size 100.”
6. The literary sources for the formulas used in the study have been identified, providing proper attribution and acknowledging previous research in the field.
All sources were clarified for every formula in the paper and also, we cited our previous experiences using those formulas.
Main source for these formulas were [1-3]:
- Oppenheim, A.V.; Schafer, R.W. Discrete-time signal processing 3rd ed.; Perason 2010.
- Xie, H.; Wang, Z. Mean frequency derived via Hilbert-Huang transform with application to fatigue EMG signal analysis. Computer methods and programs in biomedicine 2006, 82, 114-120, doi:10.1016/j.cmpb.2006.02.009.
- Chen, S.J.; Peng, C.J.; Chen, Y.C.; Hwang, Y.R.; Lai, Y.S.; Fan, S.Z.; Jen, K.K. Comparison of FFT and marginal spectra of EEG using empirical mode decomposition to monitor anesthesia. Computer methods and programs in biomedicine 2016, 137, 77-85, doi:10.1016/j.cmpb.2016.08.024.
Main previous experiences were[4,5]:
Arrufat-Pié, E.; Estévez-Báez, M.; Estévez-Carreras, J.M.; Machado-Curbelo, C.; Leisman, G.; Beltrán, C. Comparison between traditional fast Fourier transform and marginal spectra using the Hilbert–Huang transform method for the broadband spectral analysis of the electroencephalogram in healthy humans. 2021, 3, e12367, doi:https://doi.org/10.1002/eng2.12367.
Arrufat-Pie, E. Application of the Hilbert-Huang Method to the development of a platform for quantitative analysis of the Electroencephalographic signal. 2019.
7. Figure 1 is the basic illustration of the global signal processing flow. It should appear at the beginning of the methods, not in the last paragraph. This should be a guideline on how to read the methods in this document. 7. Figure 1, illustrating the global signal processing flow, has been moved to the beginning of the methods section as a guideline for readers, facilitating their understanding of the methods described.The figure was located at the of the section 2 as a guideline for the entire section. The figure also was improved adding “Hz” units to the first box in the illustration for a better understanding.
8. Section Results: Results are not disclosed comprehensively and correctly. Ln284-302 apper like Discussion. 8. The Results section has been modified to ensure comprehensive and correct disclosure of the findings. Specifically, paragraphs that were previously included in the Results section but were more suitable for the Discussion section have been moved accordingly. This section was modified as follow moving part of the ideas to the discussion: “The method of grand averages showed that the overlapping of spectral content be-tween IMFs, observed in Figures 4A and B for an individual, was also found for the whole group of healthy volunteers (see Figures 4 B, C). The grand average of the distributions of the instantaneous frequencies of the IMFs showed in general a normal distribution for the EEG leads. An illustration of this fact is shown related to the results obtained for the EEG lead C3 (Figure 4A). Because of the constant properties of the discrete frequencies of the FFT method only the spectra of the IMFs were depicted in Figures 4 B and C. The approach based on the HHT also showed the presence of similar frequencies in different IMFs, but the degree of the overlapping appears to be less marked than the observed FFT spectra of the IMFs for the same EEG lead”9. Results: missing measurement units of IMF numerical values cited in the text. 9. The measurement units of IMF numerical values cited in the text have been added for clarity and consistency. This paragraph in the methods section was clarified adding “Hz” units to a better understanding: “The grand average ranges of the mWf considering both methods values depicted in Figure 6 revealed that the IMF1 showed values from 40.3 - 52.4 Hz, for the IMF2 values of 23.84 – 31.03 Hz, for the IMF5 values from 3.56 – 5.04 Hz, the IMF6 from 1.72 – 2.77 Hz, but for the IMF3 values were observed from 10.01 – 15.79 Hz, while for the IMF4 the observed values were from 6.74 – 10.71 Hz and both frequency ranges (IMF3 and IMF4) were slightly overlapped.”
Also, the units were clarified in Table 1
10. Figure 5: The symbol '*' cannot be seen, therefore, significant differences cannot be tracked. 10. Figure 5 has been split into two separate figures (Figures 5 and 6) to enhance visibility and facilitate tracking of significant differences.11. The title can be further revised to improve the understanding of the importance and meaning of the study, as well as to follow a proper English style. 11. The title of the paper has been revised to better convey the importance and meaning of the study, while adhering to proper English style.
The title now reads, “Beyond frequency band constraints in EEG analysis: The role of the mode decomposition in pushing the boundaries.”
